# Decoupled Data Augmentation for Improving Image Classification

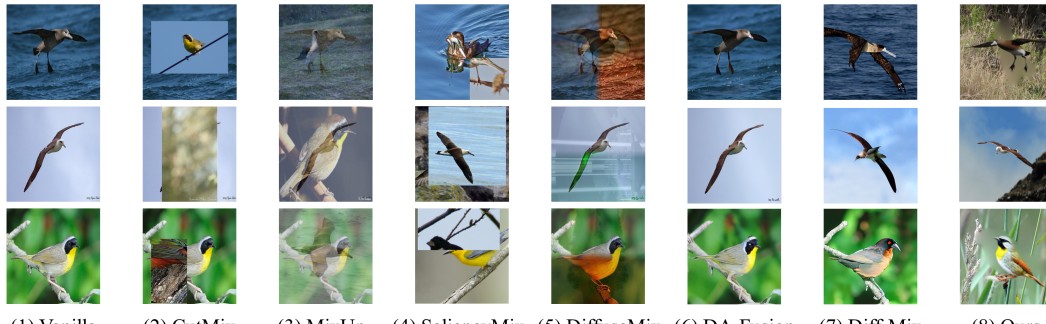

|  |  |  |  |  |  |  |  |
|---|---|---|---|---|---|---|---|
| (1) Vanilla | (2) CutMix | (3) MixUp | (4) SaliencyMix | (5) DiffuseMix | (6) DA-Fusion | (7) Diff-Mix | (8) Ours |

Figure 1: Visualization of different data augmentation methods. **Column 2-4:** CutMix, MixUp, and SaliencyMix interpolate two images at the pixel level. **Column 5-7:** DiffuseMix, DA-Fusion, and Diff-Mix utilize generative models to make semantic modifications to the input image. **Column 8:** Our proposed method, De-DA, fuses image-mixing with generative data augmentation. De-DA edits the class-dependent part of one image using a generative model, then mixes it with another image's class-independent part to create a realistic and diverse image.

## Abstract

Recent advancements in image mixing and generative data augmentation have shown promise in enhancing image classification. However, these techniques face the challenge of balancing semantic fidelity with diversity. Specifically, image mixing involves interpolating two images to create a new one, but this pixel-level interpolation can compromise fidelity. Generative augmentation uses text-to-image generative models to synthesize or modify images, often limiting diversity to avoid generating out-of-distribution data that potentially affects accuracy. We propose that this fidelity-diversity dilemma partially stems from the whole-image paradigm of existing methods. Since an image comprises the class-dependent part (CDP) and the class-independent part (CIP), where each part has fundamentally different impacts on the image's fidelity, treating different parts uniformly can therefore be misleading. To address this fidelity-diversity dilemma, we introduce Decoupled Data Augmentation (De-DA), which resolves the dilemma by separating images into CDPs and CIPs and handling them adaptively. To maintain fidelity, we use generative models to modify real CDPs under controlled conditions, preserving semantic consistency. To enhance diversity, we replace the image's CIP with inter-class variants, creating diverse CDP-CIP combinations. Additionally, we implement an online randomized combination strategy during training to generate numerous distinct CDP-CIP combinations cost-effectively. Comprehensive empirical evaluations validate the effectiveness of our method.

## 1 Introduction

Data augmentation is extensively employed to enhance neural network performance. Traditional data augmentation, such as random shifting, cropping, and rotation, are widely used due to their simplicity and effectiveness, becoming standard practice in nearly all training algorithms. Recently, two innovative types of data augmentation have shown potential for improving image classification:

- **Image-Mixing Data Augmentation.** Generate augmented images by integrating two or more randomly picked natural images at the pixel or feature level, creating virtual data between classes. The online combination paradigm allows for the efficient production of many images with extensive pixel-level variations at a low cost, yet the images often look unrealistic and face fidelity problems, as noted by (Kang & Kim, 2023; Islam et al., 2024).

- **Generative Data Augmentation.** This method leverages generative models to create images using prompts generated manually or via textual inversion to align with class labels. However, as noted by (Islam et al., 2024), this method is not yet mature for data-rich learning scenarios. Crafting prompts that ensure model-generated images match the actual data distribution is difficult, requiring expert knowledge to describe class objects and challenges in capturing the dataset's style. Additionally, textual inversion often leads to limited image diversity due to information loss, reducing the diversity of the generated images, as mentioned by (Wang et al., 2024). Both forms of prompt guidance encounter issues of misalignment or limited variation, resulting in limited performance improvements.

Readers can refer to Figure 1 for examples of various data augmentation methods. It is evident that a trade-off exists between semantic fidelity and diversity in these methods. Naturally, the question arises: *'How can semantic fidelity be preserved while simultaneously enhancing diversity?'*

The prevailing practice of treating images as indivisible units in existing data augmentation methods presents a fundamental obstacle to achieving both fidelity and diversity. This whole-image paradigm, while enriching diversity, often results in excessive and detrimental variations to class-dependent objects, severely compromising fidelity. In contrast, viewing images from a disentangled perspective could alleviate this challenge by applying distinct strategies to class-dependent parts and class-independent parts: a conservative strategy on CDPs to maintain fidelity and an aggressive strategy on CIPs to enhance diversity.

Based on this insight, we propose a novel data augmentation framework, Decoupled Data Augmentation (De-DA), which addresses the fidelity-diversity dilemma through a decoupling strategy. Specifically, we first separate images into class-dependent parts (CDPs) and class-independent parts (CIPs) using SAM (Kirillov et al., 2023), and then tailor our adaptive strategies for respective parts according to their distinct characteristics. To preserve semantic fidelity, we use class identifiers derived from intra-class CDPs as conditions to edit real CDPs with controlled strength, elaborately varying them while preserving their semantic consistency. To encourage diversity, we replace the original CIP of the images with a random CIP sampled from an inter-class image. Furthermore, we adopt an online randomized combination strategy, pairing one CDP (real or synthetic) with one CIP (cross-class real CIPs) at random positions and transformations to provide the model with various combinations during the training stage, further enhancing diversity. In summary, both conservatively translated CDPs and real CIPs align with the actual data, ensuring that the generated images maintain fidelity, while the semantic edits on CDPs and diverse CDP-CIP combinations significantly enrich variety.

Compared to previous image-mixing methods, De-DA fuses CDPs and CIPs at the semantic level rather than the pixel level, thereby enhancing fidelity. De-DA also distinguishes itself from other generative methods via applying textual inversion (Gal et al., 2022) and SDEdit (Meng et al., 2021) to isolated CDPs instead of the entire image, thus avoiding the negative effects of noisy information in the image. Furthermore, De-DA's decouple-and-combine paradigm enables the production of more images at a lower cost than prior generative methods. Our contributions include:

- De-DA shows a solution to the fidelity-diversity dilemma in previous data augmentation methods by decoupling images into class-dependent parts and class-independent parts and managing these parts adaptively.

- To our knowledge, we are the first to apply textual inversion and SDEdit to isolated CDPs instead of entire images in the field of data augmentation, which minimizes the negative impact from the noisy information in the images. Additionally, we propose truncated-timestep textual inversion to reduce the computational burden, enhancing practicability.

- Extensive experiments on domain-specific classification, multi-label classification, and data-scarce learning scenarios comprehensively validate the effectiveness of De-DA.

Table 1: Comparing data augmentation methods on fidelity and diversity.

| | Image-Mixing | | Generative | | | Image-Mixing + Generative | |
| --- | --- | --- | --- | --- | --- | --- | --- |
| | Mixup 2018 | CutMix 2019 | Real-Guidance 2023 | DA-Fusion 2024 | Diff-Mix 2024 | DiffuseMix 2024 | **De-DA (ours)** |
| Mixing | Pixel-Wise | Patch-Wise | — | — | — | Mask-Wise | Semantic-Wise |
| Prompt | — | — | Label Description | Derived from Intra-Class Images | Derived from Inter-Class Images | Style Prompt | Derived from Intra-Class CDPs |
| Fidelity | **Low** | **Low** | **High** | **High** | **Medium** | **High** | **High** |
| Diversity | **High** | **High** | **Medium** | **Low** | **High** | **Medium** | **High** |

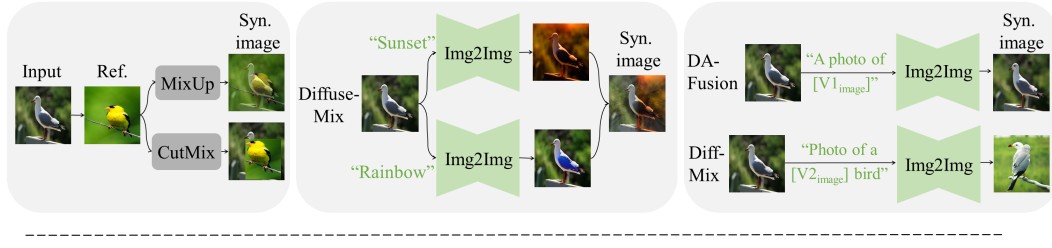

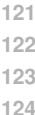
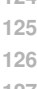

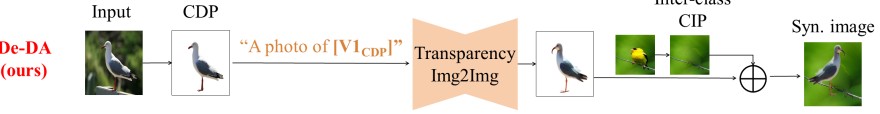

Figure 2: Illustration of the mechanisms of different data augmentation methods. **Row 1:** Image-mixing methods, such as Mixup (Zhang et al., 2018) and CutMix (Yun et al., 2019) create mixed images through pixel-level interpolation. DiffuseMix (Islam et al., 2024) uses style prompts (e.g., "Sunset") to transform input images, generating varied-style images which are then concatenated to form a hybrid image. DA-Fusion (Trabucco et al., 2024) uses the intra-class identifier $V1_{image}$, while Diff-Mix (Wang et al., 2024) employs an another class's identifier $V2_{image}$ to translate natural images with SDEdit, but these methods face issues of limited variety or constrained fidelity. **Row 2:** Our proposed De-DA maintains fidelity by editing CDPs conditioned with $V1_{CDP}$ through a transparency image-to-image diffusion pipeline which is specifically designed for handling transparent images. It also enhances diversity by replacing CIPs and applying random transformations to CDPs, resulting in faithful and diverse images.

## 2 RELATED WORK

Image-mixing and generative data augmentation methods are two approaches akin to De-DA. Table 1 offers an overview of prominent image-mixing and generative data augmentation methods, with Figure 2 depicting their mechanisms.

**Image-Mixing Data Augmentation.** Image mixing is a non-generative data augmentation technique used during training to provide classifiers with numerous mixed images, which helps smooth decision boundaries and enhance image classification (Zhang et al., 2018). Early methods, such as Mixup (Zhang et al., 2018) and CutMix (Yun et al., 2019), create new images by linearly combining two images at the pixel or patch level. However, this mixing can compromise the semantic integrity of class-specific objects. To address this, advanced approaches like SaliencyMix (Uddin et al., 2020), SnapMix (Huang et al., 2021), PuzzleMix (Kim et al., 2020a), CoMixup (Kim et al., 2020b), and GuidedMixup (Kang & Kim, 2023) use saliency maps to ensure important regions are preserved. Despite this guidance, class-specific objects may still be distorted, producing virtual images that deviate significantly from the actual data distribution, resulting in limited semantic fidelity. In contrast, De-DA addresses this issue by combining CDPs and CIPs at the semantic level, rather than at the pixel level, to preserve fidelity.

**Generative Data Augmentation.** Generative data augmentation leverages advanced text-to-image models to create new images. Initial research (He et al., 2023) demonstrates that text-to-image diffusion can generate synthetic data that effectively enhances classification performance in data-scarce scenarios, particularly when conditioned on detailed class descriptions. Azizi et al. (2023) improved generation quality by fine-tuning diffusion models on ImageNet, leading to better classification per-

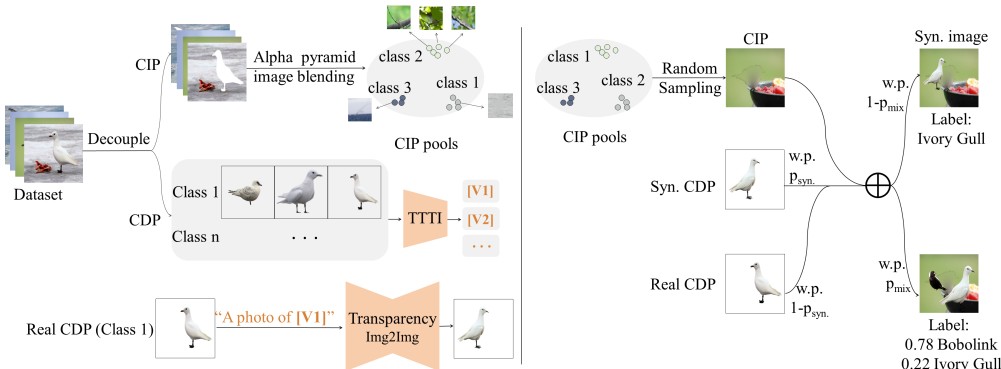

Figure 3: The pipeline of De-DA. **Left:** (1) Images are decoupled into CDPs and CIPs. Missing regions in each CIP are inpainted, creating a pool of inter-class inpainted CIPs. (2) **Truncated-Timestep Textual Inversion** (TTTI) are applied to the real CDPs to efficiently learn the class-specific identifiers $V_1, V_2, \ldots$ for each class. These identifiers are then used to semantically modify real CDPs into new synthetic CDPs. **Right:** (3) CDPs and CIPs are combined by pairing a real or synthetic CDP (with probability $1 - p_{syn}$ or $p_{syn}$) with a randomly selected CIP to create a new image. With probability $p_{mix}$, an inter-class CDP is added to generate a mixed-CDPs image.

formance. Bansal & Grover (2023); Yuan et al. (2022) utilized style prompts to synthesize images for improving domain adaptation. Subsequent studies explored various prompt enhancement techniques. For instance, Li et al. (2024) added image captions to prompts for better distribution alignment, while Yu et al. (2023) employed large language models to create diverse and detailed prompts at scale. However, these methods often struggle to produce in-distribution images due to the black-box nature of generative models and challenges in accurately describing the abstract characteristics of datasets. Consequently, they do not significantly enhance domain-specific classification tasks, where similar data distributions between different classes increase the likelihood of generating images with ambiguous labels. To address this issue, researchers leverage textual inversion (Gal et al., 2022) to learn class identifiers from real images, bypassing the difficulties of prompt design. Using these learned class identifiers, DA-Fusion (Meng et al., 2021) employs SDEdit to modify real samples into new ones with controlled generation strength, ensuring semantic consistency but limited variation. Diff-Mix (Wang et al., 2024) augments images with inter-class personalized identifiers to vary the images' backgrounds while maintaining a faithful foreground. However, there is a non-negligible probability of producing unexpected images with unfaithful foregrounds and unchanged backgrounds, necessitating the use of CLIP as a filter to remove problematic samples. DiffuseMix (Islam et al., 2024) uses manually crafted style prompts, such as "Sunset," to vary input images and concatenate two varied-style images into a hybrid image. However, the diversity in DiffuseMix is primarily reflected images' style, with limited variation in images' semantic content. We conclude that existing generative data augmentation methods often sacrifice either diversity or fidelity. De-DA mitigates this issue through a decoupling strategy, effectively controlling CIP to diversify and CDP to maintain fidelity.

## 3 DECOUPLED DATA AUGMENTATION

De-DA is a framework designed to address the fidelity-diversity trade-off through a decoupling strategy. As illustrated in Figure 3, it initially separates class-dependent parts (CDPs) and class-independent parts (CIPs) using SAM, which forms the foundation of De-DA. De-DA employs class identifiers derived from intra-class CDPs to conditionally edit real CDPs, then pairs a real or synthetic CDP with randomly selected CIPs to create new images.

**Decoupling Images into CDPs and CIPs.** The initial phase of De-DA involves separating training samples into class-dependent parts and class-independent parts, as the basement of our De-DA. Practically, we utilize Lang-SAM (Kirillov et al., 2023)[1], an off-the-shelf, prompt-based segmen-

---

[1]Lang-SAM (https://github.com/luca-medeiros/lang-segment-anything) is a prompt-guided segmentation tool based on GroundingDINO (Zuwei Long, 2023) and SAM (Kirillov et al., 2023).

tation tool, to obtain segmentation masks for class-dependent parts using domain or class name prompts (e.g., "bird" for CUB-200-2011). If multiple CDP masks are generated in one image, they are aggregated into a single mask to ensure complete coverage of the CDP. Masked regions are labeled as CDPs, while remaining image portions are classified as CIPs. We apply alpha pyramid image blending to fill the missing areas in the segmented CIPs.

**Conservative Generation of CDP.** Following prior research (Trabucco et al., 2024; Zhou et al., 2023), we use textual inversion to derive identifiers for each class and employ SDEdit to transform natural images conditioned on these prompts. Unlike previous methods, our approach applies textual inversion and SDEdit solely to the class-dependent parts (CDPs) rather than the entire image. This strategy addresses two critical issues: (1) Learning class prompts from CDPs ensures that the derived concept accurately corresponds to class-specific objects. (2) Applying SDEdit to CDPs prevents interference from class-independent parts, enhancing SDEdit's performance. This contrast is shown in Figure 4. However, applying textual inversion and SDEdit to CDPs is challenging, as traditional methods are only designed for RGB images. To accommodate transparent CDPs, we employ LayerDiffuse (Zhang & Agrawala, 2024), which equips diffusion models with a dedicated transparency encoder and decoder capable of encoding the alpha channel into latents and decoding latents into RGBA images. Specifically, our **transparency image-to-image pipeline** operates as follows: we first add noise $\epsilon \sim \mathcal{N}(0, 1)$ to real CDPs (indicated by $x_0^{\text{ref}}$) at timestep $\lfloor T_s \rfloor$, where $s \in [0, 1]$ indicates the generation strength ($s = 0$ refers no editing and $s = 1.0$ indicates generation from scratch), followed by denoising:

$$x_{\lfloor S_{T_s} \rfloor} = \sqrt{\tilde{\alpha}_{\lfloor S_{T_s} \rfloor}} x_0^{\text{ref}} + \sqrt{1 - \tilde{\alpha}_{\lfloor S_{T_s} \rfloor}} \epsilon \tag{1}$$

Denoise $x_{\lfloor S_{T_s} \rfloor}$ using LayerDiffuse reverse diffusion conditioned on the learned identifier $\mathbf{V}_{\text{CDP}}$ (we will discuss later), starting from the timestep $\lfloor T_s \rfloor$ to 0, yielding the final edited CDP $x_0$.

$$x_{t-1} = x_t - \epsilon_\theta \left( x_t, t, \mathbf{V}_{\text{CDP}} \right), \quad t = \lfloor S_{T_s} \rfloor, \ldots, 1 \tag{2}$$

Here, $\mathbf{V}_{\text{CDP}}$ is the class identifier derived from each class's real CDPs using textual inversion. To alleviate the computational cost of textual inversion, we apply **truncated-timestep textual inversion** tailored for SDEdit. In this method, the prompts are trained only on the timestep from 0 to $\lfloor S_{T_s} \rfloor$ instead of all timesteps, promoting quicker convergence. Formally, truncated-timestep textual inversion learns $\mathbf{V}_{\text{CDP}}$ by

$$\mathbf{V}_{\text{CDP}} = \arg \min_c \mathbb{E}_{t \in [0, \lfloor S_{T_s} \rfloor]} \left[ \| \epsilon - \epsilon_\theta(x_t, c, t) \| \right] \tag{3}$$

**Inter-class Random Sampling of Class-Independent Parts.** Our approach to handling Class-Independent Parts (CIPs) derives from observations of real datasets. Specifically, real datasets exhibit significant intra-class uniformity but restricted cross-class diversity. For example, in the CUB-200-2011 dataset (Wah et al., 2011), 90% of Common Yellowthroat images feature branches in the background. Albatross images often show water surfaces, while Jaeger images typically capture them in flight. Based on these observations, we propose an intra-class CIP sampling strategy rather than generating new CIPs. This method sufficiently enhances the CIP diversity of synthetic images and has proven effective in our experiments. The inter-class CIP replacement strategy offers two main advantages: (1) it is computationally efficient and avoids producing out-of-distribution CIPs, which can occur with generative data augmentation; and (2) CIP replacement can generate numerous images with the same CDPs but different CIPs, thereby enabling the trained model to better focus on critical regions of the images, which is validated in experiments 4.2.

**Online Randomized Combination.** During the training phase, we combine CDPs and CIPs by selecting a CDP and a CIP at random. The CDP is randomly resized and pasted onto a random position on the CIP, with specific random transformations applied to generate a new image. This random placement enables the model to learn position-independent features effectively. Additionally, two different CDPs are occasionally mixed with one CIP to create new multi-label samples, resulting in semantic interpolations between two classes. The weight of each label is proportional to the pixel area of the respective CDPs, following the implementation of CutMix (Yun et al., 2019). Compared to interpolations created by image-mixing data augmentation, our semantic interpolations appear more realistic and maintain semantic fidelity. Experiments (Section 4.3) confirm that the randomized combination and CDP mixing strategies could lead to performance improvements. Besides

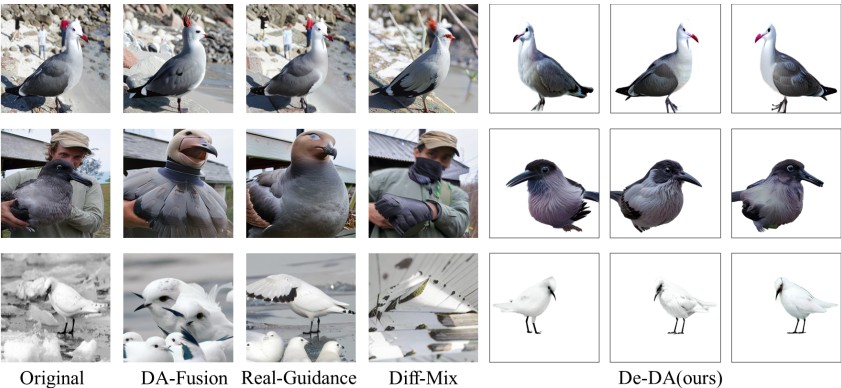

Figure 4: Examples illustrate the differences between applying SDEdit to the entire images and the pure CDPs. We observe that in generative methods, the background can negatively affect the performance of SDEdit. **Row 1:** The generative model misinterprets a person in red clothing in the background as the bird's crest. **Row 2:** A person in the background is mistakenly integrated into the bird during SDEdit, reducing the fidelity of the translated image. **Row 3:** Ice in the background is misrepresented as birds. In contrast, the right three columns showcase images generated by De-DA. De-DA involves applying textual inversion and SDEdit to isolated CDPs, facilitating modifications to avian features like feathers, eyes, and legs without altering their labels. By focusing on isolated CDPs, De-DA effectively mitigates the influence of background noise on the translation process.

superior generation quality, the decouple-and-combine prototype of De-DA can produce a substantially larger number of images with high efficiency compared to generative methods. Specifically, suppose each class consists of $M$ training samples. If we generate $K$ synthetic CDPs for each real CDP, we can obtain a total of $CM(1 + K)$ different CDP-CIP combinations ($C$ is the number of classes) for that image, where $1 + K$ is the total number of real and synthetic CDPs derived from that real CDP and $CM$ is the total number of different CIPs.

## 4 EXPERIMENTS

In this section, we comprehensively analyze De-DA by answering the following questions:

**Q1:** Can De-DA outperform other methods in conventional classification tasks?

**Q2:** Can De-DA still surpass other methods in various settings such as data-scarce scenarios?

**Q3:** How do the modules in our approach and the hyperparameters affect our method's performance?

To answer **Q1**, in Section 4.1, we compare De-DA to peer methods across different domain-specific datasets. In Section 4.2, we address **Q2** by examining its performance in data-scarce scenarios, multi-label classification, and a replaced-background dataset, demonstrating the performance gains of De-DA in various contexts. For **Q3**, we conduct extensive ablation studies on each module and hyperparameter to assess their impacts and explain our chosen settings in Section 4.3.

### 4.1 COMPARISON ON CONVENTIONAL CLASSIFICATION

**Experimental Setting.** We tested data augmentation methods on three classical domain-specific datasets: CUB-200-2011 (Wah et al., 2011), Aircraft (Maji et al., 2013), and Stanford Cars (Krause et al., 2013), following the experimental settings of DiffuseMix (Wang et al., 2024) and Diff-Mix (Wang et al., 2024). Experiments are conducted using three smaller models—ResNet-18 (He et al., 2016), ResNet-50 (He et al., 2016), and DenseNet121 (Huang et al., 2017)—as well as a large pretrained model, ViT-B/16 (Dosovitskiy et al., 2021). To ensure fairness, we adhere to prior work for our training implementations. For the small models, we followed the GuidedMix implementation (Kang & Kim, 2023), training from scratch with cross-entropy loss.[2] For ViT-B/16, we follow Diff-Mix, fine-tuning the ViT model with label smoothing loss. Unless otherwise specified, in De-DA, we set the expansion multiplier for each real CDP to 3. The generation strength for textual inversion

---

[2]Our results might differ from Diff-Mix for small models (ResNet-18, ResNet-50) because we trained from scratch, whereas Diff-Mix began with a pretrained model.

Table 2: (Training from scratch) Conventional classification on domain-specific dataset. We bolden the highest and underline the second highest.

| | Resnet-18@448 | | | Resnet-50@448 | | | DenseNet-121@448 | | | |
|---|---|---|---|---|---|---|---|---|---|---|
| Method | CUB | Aircraft | Car | CUB | Aircraft | Car | CUB | Aircraft | Car | Avg. |
| Vanilla(CVPR'2016) | 72.78 | 72.52 | 89.44 | 72.54 | 71.53 | 91.32 | 78.20 | 76.09 | 90.60 | 79.45 |
| Mixup (ICLR'2018) | 74.73 | 73.12 | 88.41 | 75.96 | 74.17 | 90.04 | 79.41 | 78.94 | 91.36 | 80.68 (+1.23) |
| CutMix (ICCV'2019) | 70.35 | 72.91 | 89.39 | 74.77 | 73.51 | 90.93 | 79.93 | 78.43 | 91.74 | 80.33 (+0.88) |
| SaliencyMix(ICLR'2020) | 70.62 | 70.36 | 87.71 | 72.92 | 73.54 | 90.52 | 78.72 | 78.25 | 91.67 | 79.37 (-0.08) |
| Co-Mixup (ICLR'2020b) | 77.25 | 72.22 | 89.44 | 79.41 | 75.70 | 90.81 | 80.89 | 78.91 | 90.44 | 81.68 (+2.23) |
| Guided-AP(AAAI'2023) | 77.77 | 75.64 | 89.62 | 78.65 | 73.48 | 89.35 | 78.15 | 78.73 | 90.97 | 81.38 (+1.93) |
| Guided-SR(AAAI'2023) | 77.27 | 76.75 | 89.52 | 78.77 | 77.38 | 91.01 | 81.27 | 80.56 | 91.22 | 82.75 (+3.30) |
| Real-Guidance (ICLR'2023) | 67.81 | 63.07 | 84.87 | 68.54 | 66.40 | 87.63 | 75.08 | 75.76 | 91.46 | 75.62 (-3.83) |
| DiffuseMix(CVPR'2024) | 73.13 | 70.03 | 88.68 | 74.40 | 73.37 | 90.60 | 79.15 | 76.00 | 91.56 | 79.66 (+0.21) |
| DA-Fusion(ICLR'2024) | 70.30 | 64.03 | 88.17 | 72.16 | 65.68 | 89.47 | 78.49 | 71.38 | 91.21 | 76.77 (-2.68) |
| Diff-Mix (CVPR'2024) | 76.32 | 77.65 | 92.35 | 77.58 | 79.21 | 93.72 | 81.41 | 83.83 | 93.68 | 83.97 (+4.52) |
| De-DA | 80.07 | 82.06 | 92.23 | 80.82 | 84.79 | 93.04 | 83.60 | 86.77 | 93.52 | 86.32 (+6.87) |

Table 3: Accuracy of finetuning on Vit-B/16.

| Method | CUB | Aircraft | Car | Avg. |
|---|---|---|---|---|
| Vanilla(CVPR'2016) | 89.37 | 83.50 | 94.21 | 89.03 |
| CutMix(ICCV'2019) | 90.52 | 83.50 | 94.83 | 89.62 (+0.59) |
| SaliencyMix(ICLR'2020) | 89.94 | 83.24 | 93.47 | 88.88 (-0.15) |
| Co-Mixup(ICLR'2020b) | 88.81 | 82.76 | 93.12 | 88.23 (-0.80) |
| Guided-AP(AAAI'2023) | 88.65 | 82.79 | 92.99 | 88.14 (-0.89) |
| Guided-SR(AAAI'2023) | 89.80 | 84.24 | 93.56 | 86.70 (+0.17) |
| Real-Guidance(ICLR'2023) | 89.54 | 83.17 | 94.65 | 89.12 (+0.09) |
| DiffuseMix(CVPR'2024) | 89.26 | 83.29 | 93.56 | 88.70 (-0.33) |
| DA-Fusion(ICLR'2024) | 89.40 | 81.88 | 94.53 | 88.60 (-0.43) |
| Diff-Mix(CVPR'2024) | 90.05 | 84.33 | 95.09 | 89.82 (+0.79) |
| De-DA | 90.62 | 84.01 | 95.15 | 89.93 (+0.90) |

Table 4: 10-shot test accuracy on CUB-200-2011.

| Method | Resnet18 | Resnet50 | DenseNet121 | Avg. |
|---|---|---|---|---|
| Vanilla(CVPR'2016) | 30.32 | 26.86 | 38.18 | 31.79 |
| Mixup(ICLR'2018) | 34.31 | 33.00 | 34.43 | 33.91 (+2.12) |
| CutMix(ICCV'2019) | 24.96 | 22.23 | 22.90 | 23.36 (-8.42) |
| SaliencyMix(ICLR'2020) | 25.27 | 23.97 | 23.94 | 24.39 (-7.39) |
| Co-Mixup(ICLR'2020b) | 37.50 | 28.56 | 38.44 | 34.83 (+3.05) |
| Guided-AP(AAAI'2023) | 40.44 | 34.35 | 41.82 | 38.87 (+7.08) |
| Guided-SR(AAAI'2023) | 38.18 | 36.71 | 41.16 | 38.68 (+6.90) |
| Real-Guidance(ICLR'2023) | 24.56 | 23.82 | 34.98 | 27.79 (-4.00) |
| DiffuseMix(CVPR'2024) | 36.05 | 33.97 | 45.27 | 38.43 (+6.64) |
| DA-Fusion(ICLR'2024) | 26.67 | 25.16 | 34.78 | 28.87 (-2.92) |
| Diff-Mix(CVPR'2024) | 45.75 | 38.79 | 48.42 | 44.32 (+12.53) |
| De-DA | 54.52 | 49.53 | 56.97 | 53.67 (+21.88) |

Table 5: Classification result on out-of-distribution dataset Waterbird Sagawa* et al. (2020), each image is crafted by combine CUB-200-2011's foregrounds with the background from Places Zhou et al. (2017). Higher accuracy represents higher background robustness.

| Method | (Waterbird, Water) | (Waterbird, Land) | (Landbird, Land) | (Landbird, Water) | Avg. |
|---|---|---|---|---|---|
| Vanilla(CVPR'2016) | 59.50 | 56.70 | 73.48 | 73.97 | 70.19 |
| Mixup(ICLR'2018) | 66.67 | 61.37 | 74.28 | 75.52 | 72.52 (+2.33) |
| CutMix(ICCV'2019) | 62.46 | 60.12 | 73.39 | 74.72 | 71.23 (+1.04) |
| Real-Guidance (ICLR'2023) | 61.06 | 56.08 | 70.73 | 71.40 | 68.29 (-1.9) |
| DiffuseMix(CVPR'2024) | 63.08 | 57.48 | 71.35 | 74.46 | 70.11 (-0.08) |
| DA-Fusion(ICLR'2024) | 60.90 | 58.10 | 72.94 | 72.77 | 69.90 (-0.29) |
| Diff-Mix(CVPR'2024) | 63.83 | 63.24 | 75.64 | 74.36 | 72.47 (+2.28) |
| De-DA | 67.72 | 67.32 | 78.40 | 78.85 | 76.17 (+5.98) |

and SDEdit is fixed at $s = 0.4$. During training with De-DA, nautral images are replaced with augmented data with a probability $p_{aug} = 0.5$. For CDP-CIP combinations, the probability of using mixed CDP $p_{mix}$ is 0.5, and the synthetic CDP is used with a probability $p_{syn} = 0.25$.

**Peer Methods.** We compare De-DA with ten peer methods, including six image-mixing and four generative approaches. The image-mixing methods include: (1) Mixup (Zhang et al., 2018), which linearly combines pairs of images and their labels; (2) CutMix (Yun et al., 2019), which replaces a portion of one image with a patch from another; (3) SaliencyMix (Uddin et al., 2020); (4) Co-Mixup (Kim et al., 2020b); and (5) Guided-AP and (6) Guided-SR (Kang & Kim, 2023), which use saliency maps to guide the mixing process, alleviating the issue of corrupted class-specific objects. The generative methods include: (1) Real-Guidance (He et al., 2023), which augments the dataset using label-name guidance at a fixed low strength $s = 0.1$; (2) DiffuseMix (Islam et al., 2024), which creates hybrid images from different conditional prompts using fractal blending; (3) DA-Fusion (Trabucco et al., 2024), which augments images with identifiers learned from intra-class images at random strengths $s \in \{0.25, 0.5, 0.75, 1.0\}$. (4) Diff-Mix (Wang et al., 2024), which augments images with the identifiers learned from other classes' images at random strengths $s \in \{0.5, 0.7, 0.9\}$. Both Real-Guidance and Diff-Mix additionally make use of the large vision-language model CLIP Radford et al. (2021) to evaluate label confidence, aiding in the filtering of distorted images. We set the expansion multiplier for all augmented methods and adjust the augmentation probability $p_{aug}$ according to each method's recommendation.

**Comparison on Conventional Classification.** Table 2 presents the test accuracy of various augmentation strategies across three domain-specific datasets. All methods use an input resolution of

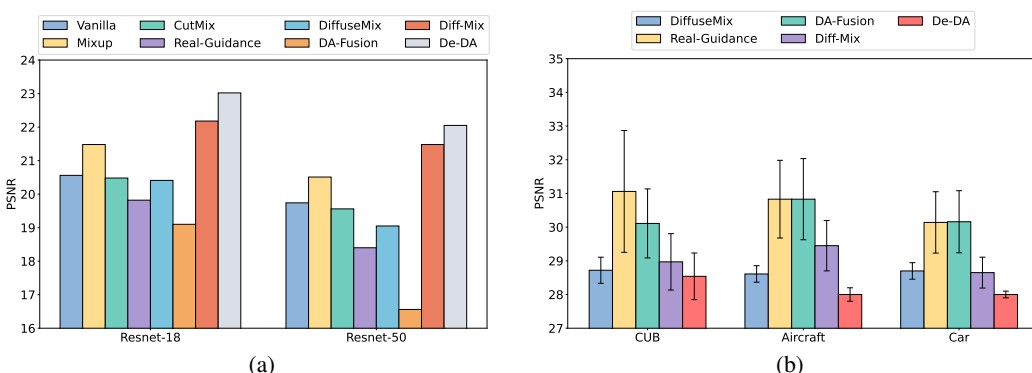

Figure 5: (a) Comparison on multi-label classification. (b) Comparison on diversity by PSNR.

448 and are trained from scratch. The results indicate the following: (1) De-DA consistently outperforms other data augmentation methods on CUB-200-2011 and Aircraft by a significant margin. Specifically, De-DA achieves test accuracies of 80.07% and 82.06%, exceeding the second-highest accuracy by notable margins of 2.3% and 4.41%, respectively, with ResNet-18. (2) De-DA does not surpass Diff-Mix in test accuracy on Stanford Cars. We hypothesize this is due to the class-specific object occupying most of the image area in Stanford Cars, making the CIP replacement strategy less effective for performance gains in this dataset. (3) Generative augmentation methods like DA-Fusion and Real-Guidance do not enhance accuracy, suggesting that the standard use of diffusion for image editing is limited in its ability to improve accuracy due to restricted diversity.

**Comparison on Fine-Tuning on Large Models.** Table 3 reports the accuracy of different methods on large model fine-tuning with an input resolution of 384. The results indicate: (1) De-DA consistently demonstrates superior performance compared to other methods. (2) The improvement of De-DA over the second-highest method is less pronounced than when training from scratch.

## 4.2 COMPARISONS ON VARIOUS TASKS

To evaluate performance in data-scarce scenarios, we create a version of the CUB-200-2011 dataset by randomly selecting 10 images per class, following the settings of DiffuseMix (Islam et al., 2024). To assess how data augmentation aids in learning background-robust features, we test accuracy on the Waterbird dataset, which combines bird foregrounds from CUB-200-2011 with backgrounds from the Places dataset (Zhou et al., 2017). Here, (Waterbird, Water) indicates (the type of bird, the type of background). We further compare De-DA to other methods on the multi-label classification dataset Pascal (Everingham et al., 2010) to validate its performance in improving multi-label classification. Additionally, we demonstrate that De-DA is compatible with other data augmentation techniques, such as RandAugment(Cubuk et al., 2020).

**Comparison in Data-Scarce Scenarios.** The results on the data-scarce CUB-200-2011 dataset are shown in Table 4. We observe that De-DA significantly outperforms all other methods, achieving an accuracy of 54.52% on ResNet-18, which is 8.77% higher than the second-best method. This remarkable improvement is due to the substantially larger number of augmented images generated by De-DA's online combination strategy, which compensates for the lack of data. Furthermore, compared to image-mixing methods that also produce a large number of mixed images, De-DA shows significant improvement, demonstrating that the images generated by De-DA are much more effective due to their high diversity and fidelity.

**Comparison on Background Robustness.** Table 5 presents the experimental results on the Waterbird dataset, which evaluates the model's robustness against background replacement, rather than relying on the background. De-DA clearly outperforms other methods in all categories, achieving an average improvement of 5.98%, which is 3.70% higher than the second-best method, Diff-Mix. This validates our earlier statement that De-DA helps the model learn CIP-independent features, enabling the model to focus on the class-specific object for classification.

**Comparison on Multi-Label Classification.** Figure 5a compares different methods on a multi-label classification. De-DA achieves 23.02% on ResNet-18 and 22.05% on ResNet-50, surpassing

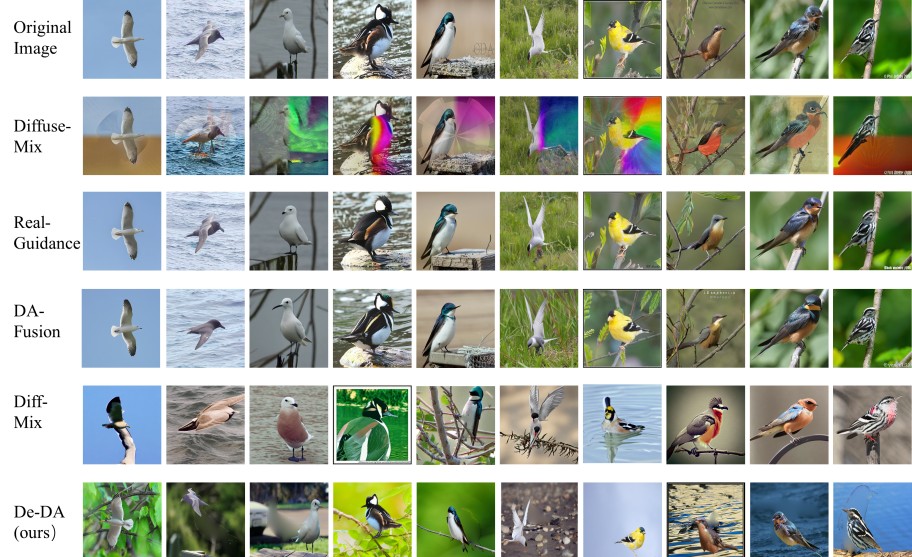

Figure 6: Visual examples of different generative data augmentation methods.

all other methods by a non-trivial margin, demonstrating that De-DA can effectively improve multi-label classification through mixed-CDPs augmented samples.

**Comparing Diversity of Generative Data Augmentation.** Figure 5b quantitatively compares De-DA to other generative methods using Peak Signal-to-Noise Ratio (PSNR). A lower PSNR value indicates higher diversity. The results show that De-DA achieves greater diversity than the other methods. Figure 6 presents the examples of different generative data augmentation methods, validating our aforementioned statement. Specifically, we observe that (1) DiffuseMix diversifies images from a stylistic perspective rather than a semantic level. While it shows robustness to adversarial noise, it is less effective in improving test accuracy. (2) Real-Guidance slightly modifies images using SDEdit at a low strength. Although it maintains semantic consistency, it struggles with background invariance. (3) Da-Fusion has the same issue as Real-Guidance. (4) Diff-Mix uses identifiers from other classes to transform the input image, aiming to vary the background while preserving the semantic fidelity of the foreground. However, it often significantly alters the foreground greatly without effectively diversifying the background.

**Compatibility with Traditional Data Augmentation.** We evaluate the compatibility of De-DA with RandAugment (Cubuk et al., 2020), as shown in Figure 7. The results indicate that combining De-DA with RandAugment outperforms using De-DA alone, indicating that De-DA and traditional data augmentation RandAugment are two mutually reinforcing mechanisms. The result validates the extensibility of De-DA.

Figure 7: Compatibility of De-DA with RandAugment.

### 4.3 ABLATION STUDIES

**Experimental Setting.** We first evaluate the impact of each hyperparameter of De-DA on CUB-200-2011 with ResNet-18. Then, to evaluate the contribution of each component of our approach, an ablation study is conducted by incrementally adding each component. We focused on components including synthetic CDP, CIP replacement, the randomized combination and the CDP mixing technique. The experimental baseline model is ResNet-18 and the dataset is Aircraft.

**Ablation on Hyperparameters.** The impact of the De-DA's hyperparameters is shown in Figure 8. The experimental results lead to the following conclusions: (1) The performance of De-DA improves as $p_{\text{aug}}$ increases from 0.00 to 0.50, peaking at 0.5, indicating that a balanced approach of using both generated and original data is optimal. (2) The highest performance for three datasets is observed at $P_{\text{syn}} = 0.25$. Using either no synthetic CDP or only synthetic CDPs results in a performance decline. (3) We observe that CDP mixing leads to significant improvement on CUB-200-2011 at $p_{\text{mix}} = 0.25$, demonstrating that CDP mixing effectively boosts classification. (4) A

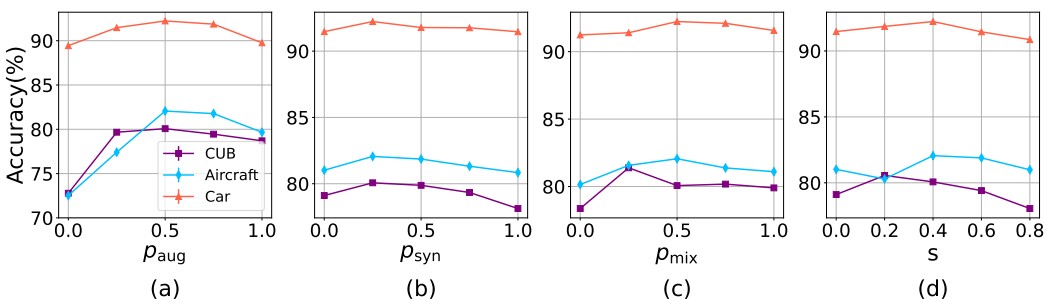

Figure 8: (a) Impact of the probability of replacing real data with augmented data $p_{\mathrm{aug}}$; (b) Impact of the probability of using edited CDPs $p_{\mathrm{syn}}$; (c) Impact of the probability of using CDP mixing $p_{\mathrm{mix}}$; (d) Impact of the generation strength $s$.

generation strength of $s = 0.4$ consistently yields improvement across the three datasets. Peak performance occurs at different strengths: Aircraft and Standford Car peak at $s = 0.4$, while CUB-200-2011 peaks at $s = 0.2$, possibly because the inter-class images in the CUB-200-2011 are more similar than in the other two datasets. However, too high a strength can result in performance decline, e.g. $s = 0.8$ achieves a lower accuracy than $s = 0.0$ on CUB-200-2011. These ablation studies explain our hyperparameter choices. The results indicate that De-DA is relatively robust to hyperparameter settings. The impact of the expansion multiplier is discussed in the appendix.

**Ablation on Each Module of De-DA.** Figure 9 shows the module ablation results. (a) represents vanilla training without data augmentation. (b) involves replacing the CDP of the original sample with a same-size synthetic CDP at the same position, with a probability $p_{syn} = 0.25$. This improves upon the baseline, validating the effectiveness of semantically edited CDPs in image classification. (c) replaces the CDP with a synthetic one using a random combination strategy that varies the position and size of the CDPs, further enhancing (b) by 1.34 %, indicating that random combination strategy do compile with CDP editing. (d) employs only the CIP replacement strategy without CDP editing or random combination, effectively boosting accuracy, which highlights the importance of CIP diversity. (e) utilizes an inter-class CIP strategy with random combination, further improving performance, indicating that CIP replacement and random combination are two mutually reinforcing mechanisms. (f) incorporates the strategies of CDP editing, inter-class CIP replacement, and random combination, achieving an accuracy of 80.15%, which surpasses the accuracies of all other peer methods. (g) is the complete version of De-DA, incorporating the CDP-mixing strategy. This integration further enhances the performance.

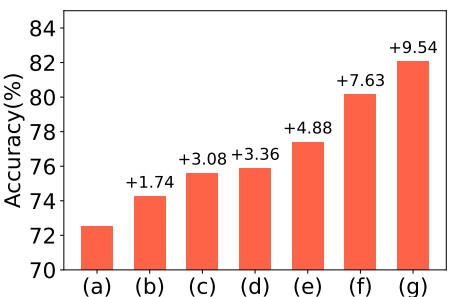

Figure 9: Ablation study on each module.

## 5 CONCLUSIONS AND FUTURE WORKS

In this work, we propose an innovative approach to address the fidelity-diversity dilemma through decoupled data augmentation (De-DA). We decouple images into class-dependent and class-independent parts, with CDP maintaining semantic fidelity and CIP enhancing diversity. The decouple-and-combine strategy of De-DA enables the production of faithful and diverse images at scale, with lower computational costs compared to generative methods. Experiments validate that De-DA effectively improves conventional classification, data-scarce classification, and multi-label classification. De-DA also helps models learn background-independent features. Future work could explore several directions: (1) decoupling images in a more fine-grained manner to improve performance in fine-grained retrieval tasks; (2) developing adaptive strategies for designing generation strength based on dataset characteristics; (3) exploring new CDP-CIP combination approach to further boost diversity.

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

# A   APPENDIX

Table 6: General hyperparameters of De-DA.

| Hyperparameter Name | Value |
|---|---|
| Segmentation prompt | CUB: "Bird"; Aircraft: "Aircraft"; Standford Car: "Car"; Pascal: label name |
| Probability of using augmented data ($p_{\text{aug}}$) | 0.5 |
| Probability of using synthetic CDP ($p_{\text{syn}}$) | 0.25 |
| Probability of applying CDP mixing ($p_{\text{mix}}$) | 0.5 |
| Generation strength ($s$) | 0.4 |
| Textual inversion token initialization | CUB: "Bird"; Aircraft: "Aircraft"; Standford Car: "Car"; Pascal: label name |
| Textual inversion batch size | 32 |
| Textual inversion learning rate | 1e-4 |
| Textual inversion training steps | 400 |
| SDEdit prompt | "a photo of a <class name>" |
| Stable diffusion guidance scale | 7.0 |
| Stable diffusion resolution (pixels) | 512 |
| Stable diffusion denoising steps | 25 |

Table 7: Training hyperparameters of De-DA.

| Architecture | Resnet-18 | Resnet-50 | DenseNet-121 | ViT-B/16 |
|---|---|---|---|---|
| Learning rate | 0.01 | 0.01 | 0.01 | 0.001 |
| Epochs | 300 | 300 | 300 | 120 |
| Batch size | 16 | 16 | 8 | 32 |

## A.1   IMPLEMENTATION DETAILS

All of our experiments are conducted on a system equipped with 96 CPU cores (Platinum 8255C @ 2.50GHz) and 8 GPU Tesla V100 cards. For doupling the images into CDPs and CIPs, we employ LangSAM [3] with prompt guided. For inpainting the missing part of For the training implementations of ResNet-18, ResNet-50, and DenseNet-121, we adhere to the official training script from GuidedMix (Kang & Kim, 2023) [4]. For the training of Vit-B/16, we followed the official implementation provided by Diff-Mix (Wang et al., 2024) [5]. Additionally, we introduce the hyperparameters related to decoupling, truncated-timestep textual inversion and SDEdit. Specific values for these hyperparameters are provided in Table 6 and Table 7.

## A.2   ABLATION STUDY

**Impact of Expansion Multiplier.** The impact of the expansion multiplier is presented in Table 8, which shows the accuarcy at different expansion multipliers $\times 1$, $\times 3$, $\times 6$, $\times 10$. De-DA achieves optimal accuracy at a multiplier of $\times 3$ for both Aircraft and Standford Car datasets, while Cub-200-2011 peaks at $\times 6$. The difference likely stems from the distinct data distributions inherent to each dataset. Notably, increasing the expansion multiplier does not necessarily improve performance, a phenomenon also observed in (Trabucco et al.,

Table 8: Ablation on the expansion multiplier for each real CDP.

| Number | CUB | Aircraft | Car |
|---|---|---|---|
| $\times 1$ | 79.68 | 81.50 | 91.91 |
| $\times 3$ | 80.07 | **82.06** | **92.23** |
| $\times 6$ | **80.18** | 81.52 | 91.92 |
| $\times 9$ | 79.89 | 80.96 | 90.85 |

2024; Wang et al., 2023). This suggests that excessive data augmentation may bias the model towards the generated data, hindering its ability to generalize effectively to real data.

---

[3] https://github.com/luca-medeiros/lang-segment-anything/

[4] https://github.com/3neutronstar/GuidedMixup/blob/main/FGVC/main.py

[5] https://github.com/Zhicaiwww/Diff-Mix/blob/master/downstream_tasks/train_hub.py

