# OpenReview forum: "Decoupled Data Augmentation for Improving Image Classification"
_ICLR.cc/2025/Conference — ICLR 2025 Conference Withdrawn Submission_

### Official Review · Reviewer_s2xK · 2024-10-24

**Soundness:** 3
**Presentation:** 2
**Contribution:** 2
**Rating:** 5
**Confidence:** 4

**Summary:**

This paper proposes a decoupled data augmentation framework for vision classification tasks, which separately augments the class-dependent part (CDP) and the class-independent part (CIP). Specifically, by employing SAM to initially separate CDP and CIP, the paper then designs a transparency img2img pipeline to augment CDP, inspired by SDEdit and LayerDiffuse, and finally generates synthetic samples by combining the generated CDPs with randomly sampled CIPs. Experiments on various datasets demonstrate the effectiveness of the proposed method.

**Strengths:**

1. The motivation is sound. The paper proposes separating foreground and background before targeted augmentation, aiming to maintain fidelity while increasing diversity. This approach is reasonable and effective, contrasting with previous methods that targeted the entire image.
2. The paper effectively leverages existing technologies like SAM and LayerDiffuse to propose a new data augmentation framework, yielding valid experimental results.
3. Extensive experimental results and open-source code suggest the method is reproducible.

**Weaknesses:**

**Major:**

1. Limited technical contribution. The paper utilizes existing techniques such as SAM, LayerDiffuse, and text inversion. While these technologies are well-utilized to produce seemingly credible results, there is a lack of novel technical contributions and design. Additionally, performance improvements gained by using SAM are unsurprising and come with increased computational costs. Would the method still work without SAM? Are there alternatives to SAM?
2. The writing needs further polish. There are confusing descriptions, particularly in the Method section's Conservative Generation of CDP. For instance, in Eq. 3, the variable $ c $ seems to refer to a text prompt, but its definition is missing. Moreover, as a critical part of augmentation, there is no experimental discussion on it. Also, if Line 233’s $\lfloor T_{s}\rfloor$ indicates a timestep, what does $\lfloor S_{T_{s}}\rfloor$ in Eq. 1 mean? $ S $ is also undefined—perhaps it means total steps? Numerous other issues are noted in the Minor weaknesses.
3. I recommend adding a Background section to provide a brief review of SDEdit and the diffusion model used, along with clear symbol definitions.
4. Experimental results are confusing. For example, the source of the experimental results is unclear—are they reproduced by the authors or cited from another paper? In Table 2, the authors label “training from scratch,” yet the vanilla results for DiffuseMix on ResNet-50@448 are 65.50, 80.29, and 85.52, compared to this paper’s 72.54, 71.53, and 91.32, which are significantly higher than those reported in DiffuseMix’s Table 14. The authors need to clarify the experimental setup and result sources. Additionally, in Table 5, results for Vanilla, CutMix, DA-Fusion, and Diff-Mix are cited from Diff-Mix, but the source for the remaining results is missing.

**Minor:**

1. Inconsistent use of \citet{} and \citep{}. Please verify correct usage in Line 058, Line 061, Line 068, and Line 346.
2. Incorrect citation. The citation for DiffuseMix in Line 315 seems wrong.
3. Incorrect Y-axis label. Why is the metric for multi-label classification in Figure 5(a) PSNR?
4. Line 731 seems incomplete. "For inpainting the missing part of For the training ..." What does this mean?????

**Questions:**

1. Long-tailed image classification is a natural data-scarce scenario to test the effectiveness of data augmentation methods. I am curious about the paper's performance on long-tailed datasets like classical Places-LT& ImageNet-LT, and CUB-LT & Flower-LT used in Diff-Mix.

2. Without SAM, could other instance segmentation methods achieve similar foreground-background separation? How significant is SAM's impact? I believe this is crucial for demonstrating the robustness of the proposed method. If it relies heavily on SAM, its effectiveness could be significantly diminished, as previous methods did not use additional tools like SAM for foreground-background separation. I am curious to see if the method is still effective under poor separation conditions.

I believe the current version does not meet the standards for acceptance. However, I acknowledge the authors' motivation and their effective use of pre-trained models to address the data augmentation problem. I am willing to increase my score if the authors can adequately address these weaknesses and questions during the rebuttal.

---

### Official Review · Reviewer_sVtp · 2024-10-30

**Soundness:** 2
**Presentation:** 3
**Contribution:** 2
**Rating:** 3
**Confidence:** 4

**Summary:**

The paper proposes De-DA, which is a framework for sample-mixing and generative data augmentation. Specifically, it decouples the frontal object and background of an image using SAM. During training, De-DA applies generative diffusion models to transform the object and paste it to an extracted background from another image. The method helps to reduce the background noise when transforming the object and shows strong experimental results on fine-grained classification tasks.

**Strengths:**

- The paper is well-written and is easy to understand

- The experimental results are promising

**Weaknesses:**

Limited Novelty. The approach of using background changes to create augmented data has been studied before. For example, InSP[1] swaps the saliency part of two images from the same class and is tested on CUB, Stanford Car, and FGVC-Aircraft datasets. Copy-paste augmentation [2] is a low-cost augmentation method that copies and pastes a random object into another image, for instance segmentation. Applying textual inversion and SDEdit to transform objects was suggested in DA-Fusion. The SDEdit and SAM models are also off-the-shelf methods proposed in previous works. The proposed idea of transforming extracted objects and pasting them into the backgrounds from other images is somewhat incremental to existing works.

De-DA proposes to extract class-dependent parts of an image using SAM. If the SAM model is not pre-trained on the target domain, it may fail to capture the class-dependent part, such as medical images. In addition, the class-dependent parts of an image are not limited to the background. For example, in gesture classification, the gesture is also a class-dependent part; in scene classification, the whole image should be considered. It seems that the proposed method cannot capture non-background class-dependent features and is limited to fine-grained object classification tasks.

Limited evaluation of the tested datasets and tasks. It appears that all three tested datasets are fine-grained classification datasets. The backgrounds in fine-grained datasets are relatively easy to transfer as the objects are mostly the same. However, this may not be the case if the object classes are semantically different. For example, swapping the background of a marine animal and a desert animal may not be appropriate. Therefore, I suggest the authors test their method on large-scale datasets with more diverse objects, e.g., the ImageNet dataset.

[1] Zhang, Lianbo, Shaoli Huang, and Wei Liu. "Intra-class part swapping for fine-grained image classification." Proceedings of the IEEE/CVF winter conference on applications of computer vision. 2021.

[2] Ghiasi, Golnaz, et al. "Simple copy-paste is a strong data augmentation method for instance segmentation." Proceedings of the IEEE/CVF conference on computer vision and pattern recognition. 2021.

**Questions:**

De-DA generates augmented images using SAM and diffusion models. How does it compare with other baselines in terms of efficiency?

The proposed method is evaluated only on fine-grained classification tasks. Some similar mixing methods are evaluated on object detection and instance segmentation tasks. For example, Copy-Paste augmentation [2] is a low-cost augmentation method that copies and pastes a random object into another image. How does De-DA compare with this baseline in instance segmentation tasks?

---

### Official Review · Reviewer_C6g3 · 2024-11-02

**Soundness:** 3
**Presentation:** 3
**Contribution:** 3
**Rating:** 6
**Confidence:** 4

**Summary:**

The paper introduces a data augmentation framework called Decoupled Data Augmentation (De-DA) to tackle the fidelity-diversity dilemma. This approach involves a decoupling strategy that separates images into class-dependent parts (CDPs) and class-independent parts (CIPs) using SAM. The method then applies text inversion and SDEdit to the CDP, and subsequently combines it with randomly selected CIPs to generate new images. This process aims to maintain semantic consistency while enhancing diversity

**Strengths:**

1. This paper introduces a simple yet effective method that balances fidelity and diversity in synthetic data. The core idea is to decouple CIPs and CDPs using an off-the-shelf segmentor, augmenting them separately and then combining them to create new samples. This approach presents a methodological innovation compared to prior generative data augmentation techniques.
2. De-DA utilizes layer-based composition to generate synthetic samples, making the synthesis process highly efficient. This is beneficial in resource-constrained settings,
3. In Table 1 and Figure 2, the authors provide a comprehensive overview of various data augmentation methods from the perspectives of diversity and fidelity, which offers valuable insights for readers aiming to understand advancements in this area.

**Weaknesses:**

1.Although the Online Randomized Combination method is efficient, it may lead to semantically unnatural compositions. For instance, birds in the synthetic samples do not always appear naturally perched on branches (see the last row of birds in Figure 6, where proper positioning on branches is rare). While semantic naturalness may not always be critical for classification, this could reduce the generalizability of synthetic data.

2.The authors should include more visualizations of De-DA results, especially showcasing foreground variations in CIP and the inpainting results for CDP. Additionally, it would be valuable to discuss some lower-quality samples and include a discussion on the limitations of the current approach.

3.Some experimental settings are overly brief. For instance, details regarding the multi-label classification implementation are missing, and the authors provide insufficient explanation on how samples are constructed in this scenario.

**Questions:**

I am uncertain about the appropriateness of using PSNR to evaluate diversity in Figure 5b. PSNR measures pixel-level deviation, meaning that a simple Mixup operation could also result in low PSNR, but it would be difficult to argue that Mixup inherently promotes high diversity. I would appreciate further clarification from the authors on this point.

---

### Official Review · Reviewer_KAQK · 2024-11-03

**Soundness:** 3
**Presentation:** 3
**Contribution:** 2
**Rating:** 5
**Confidence:** 4

**Summary:**

This paper introduces a method to decouple an image into its class-dependent part (CDP) and class-independent part (CIP), and subsequently processes these two components individually for data augmentation. Specifically, the method first uses SAM to segment out CDP and CIP. Then, it appropriately augments the CDP part using a diffusion model. After that, it randomly combines the CDP with CIP from different images to increase image diversity.

**Strengths:**

* The method of dividing images into CDP and CIP regions sounds reasonable and the experiment results show its effectiveness.
* The paper demonstrates the effectiveness of the method on different networks and datasets.

**Weaknesses:**

* The paper only conducted experiments on datasets with relatively limited data. It did not illustrate the effectiveness on larger and more general datasets. Can this method work on larger datasets like ImageNet, similar to how mixup or cutmix do?
* In the experiments comparing with RandAugment, only the results of “DE-DA” and “DE-DA + RandAugment” are provided. I believe that further results for “RandAugment only” should be included. Because it would be better to demonstrate that the “DE-DA + RandAugment” method outperforms “RandAugment only”.

**Questions:**

* Is the issue that models struggle with distinguishing background and foreground only present in scenarios with relatively limited data, or is this a problem for larger datasets?
* Can this method be regarded as distilling the knowledge from SAM and diffusion models, which have been trained on large-scale datasets?
* What is the computational complexity of the method when processing images, and is it feasible to apply it to larger datasets?

---

### Note · Authors · 2024-11-14

I have read and agree with the venue's withdrawal policy on behalf of myself and my co-authors.